# AN AI MONKEY GETS GRAPES FOR SURE – SPHERE NEURAL NETWORKS FOR RELIABLE DECISION-MAKING

## ABSTRACT

By increasing the amount and the quality of training data, we may improve the logical reasoning performances of LLMs, but they are still unreliable and struggle with simple decision-making, an ability that animals may develop without language. This paper proposes a new version of Sphere Neural Networks that embeds concepts as circles on the surface of an $n$-dimensional sphere. This new version enables the representation of the negation operator through complement circles and can achieve reliable decision-making by eliminating unsatisfiable circle configurations. Comparative experiments with supervised neural reasoning are performed by retraining Euler Net for disjunctive syllogistic reasoning, the foundation for decision-making. We demonstrate that the proposed Sphere Neural Network achieves rigorous disjunctive syllogistic reasoning as well as 15 other syllogistic-style reasoning tasks, while preserving the rigour of classical syllogistic reasoning. In contrast, an Euler Net achieving 100.00% in classic syllogistic reasoning can be trained to reach 100% accuracy in disjunctive syllogistic reasoning. However, after that, its performances dropped to 6.25% in classic syllogistic reasoning, then subsequently dropped to 75.00%, 53.57%, and 46.43% when input images had random colour, random colour and boundary thickness, or filled circles, respectively. This comparison favours the method of neural reasoning with explicit model construction and suggests seeking alternative neural methods to enhance the reliability of neural decision-making.

## 1 INTRODUCTION

Reliable decision-making is crucial in high-stakes applications. Though LLMs have achieved unprecedented success in many ways, exemplified in human-like communication (Biever, 2023), playing Go (Silver et al., 2017; Schrittwieser et al., 2020), predicting complex protein structures (Abramson et al., 2024), or weather forecasting (Soliman, 2024), they still make errors in simple reasoning (Melanie, 2023). In addition, LLMs are prone to making accurate predictions with incorrect explanations (Creswell et al., 2022; Zelikman et al., 2022; Park et al., 2024), and have not yet achieved the reliability necessary for high-stakes applications, i.e., decision-making in biomedicine (Wysocka et al., 2025). Syllogistic reasoning appears deceptively simple, for example, *Gvery member of Diseases of hemostasis pathway is a member of Disease pathway. Gene GP1BB is a member of Diseases of hemostasis pathway. ∴ Gene GP1BB is a member of Disease pathway.* However, Eisape et al. (2024) show that although LLMs perform better than average humans in syllogistic reasoning, their accuracy remains limited to around 75%, and larger models do not consistently outperform smaller ones. Lampinen et al. (2024) come to the convergent conclusions that, in abstract reasoning such as syllogism, LLMs may achieve above-chance performances in familiar situations but exhibit numerous imperfections in less familiar ones. Wysocka et al. (2025) tested LLMs in several types of generalised syllogistic reasoning, e.g., *generalised modus ponens*, *disjunctive syllogism*, in the context of high-stakes biomedicine, and found that zero-shot LLMs achieved an average accuracy between 70% on *generalised modus ponens* and 23% on *disjunctive syllogism*. Crucially, both zero-shot and few-shot LLMs demonstrated pronounced sensitivity to surface-level lexical variations.

Despite these limitations, evaluating the reasoning performance of LLMs on syllogistic tasks (and beyond) can provide insights into the origins of (human) rationality (Lampinen et al., 2024) and

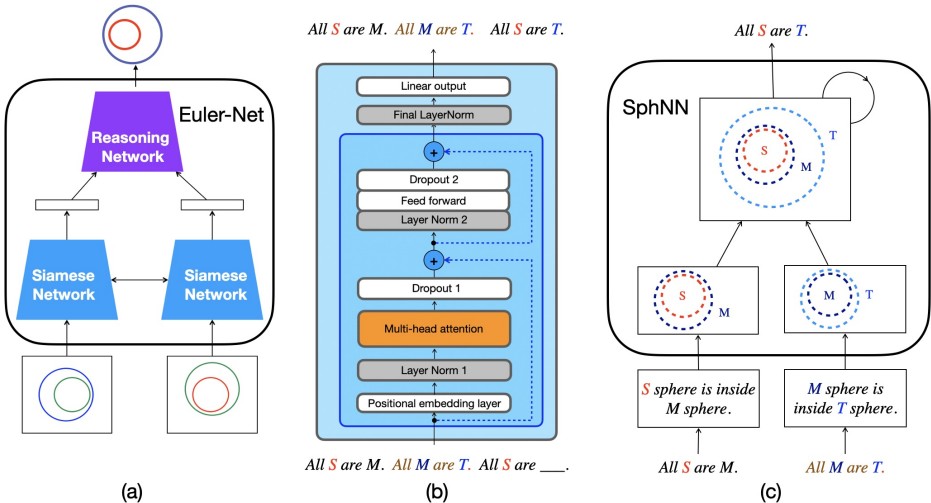

Figure 1: (a) The architecture of Euler Net. The inputs are two images, representing two premises of Aristotelian syllogistic reasoning. The green circle inside the blue circle represents "all green are blue"; the red circle inside the green circle represents "all red are green". Its output represents "all red are blue". (b) The transformer component of LLMs learns to predict the missing word. (c) The architecture of Sphere Neural Networks (SphNN) for syllogistic reasoning. SphNN transforms syllogistic statements into spatial statements between spheres, constructs a unified configuration, and draws conclusions by inspecting the constructed configuration.

help identify alternative methods for reliable neural reasoning and decision-making. LLMs acquire human-like communication and reasoning abilities by training on large-scale linguistic corpora. However, proficiency in communication does not necessarily equate to proficiency in reasoning (Fedorenko et al., 2024). On the contrary, reasoning and decision-making are abilities that do not necessarily depend on extensive language acquisition. For example, clever monkeys can get grapes through disjunctive syllogistic decision-making (Ferrigno et al., 2021), suggesting that syllogistic reasoning can be elicited through visual–spatial inputs, independent of linguistic abilities. While recent evaluations indicate that LLMs remain unable to achieve robust syllogistic reasoning, in this work, we revisit (Ferrigno et al., 2021)'s experiments to seek alternative neural architectures and neural reasoning methods. We distinguish three categories of neural networks for syllogistic reasoning as follows, and illustrate in Figure 1.

1. *Supervised networks with image inputs*. Inspired by the structural similarity between deep convolution neural networks (DNN) (He et al., 2016) and visual cortex (Yamins & DiCarlo, 2016), Euler Nets are special DNNs that perform Aristotelian syllogistic reasoning with Euler diagrammatic-styled image-inputs (Wang et al., 2018; 2020).

2. *Supervised networks with linguistic inputs*. The popular neural network architecture is the Transformer (Vaswani et al., 2017) and LLMs (Google, 2023; Touvron et al., 2023; OpenAI, 2023; Jiang et al., 2023; Mistral, 2023). The basic training method is masked word prediction (Raschka, 2024) using Transformers. Given the text *all Greeks are human. all humans are mortal. therefore, all Greeks are \_\_\_*, LLMs are trained to predict *mortal*.

3. *Neural networks that reason through explicit model construction*. Sufficient empirical experiments advocate the model theory for reasoning that reasoning is a process of constructing and inspecting mental models (Johnson-Laird & Byrne, 1991; Knauff et al., 2003; Goodwin & Johnson-Laird, 2005; Knauff, 2009; 2013). In line with the mental model theory, Sphere Neural Networks perform Aristotelian syllogistic reasoning by constructing Euler diagrams in the Euclidean or Hyperbolic space (Dong et al., 2024; 2025). They are proven to achieve the rigour of symbolic-level logical reasoning.

In this paper, we compare reasoning performances between Category 1 *supervised neural networks with image inputs* and Category 3 *neural networks through explicit model construction* (in the Ap-

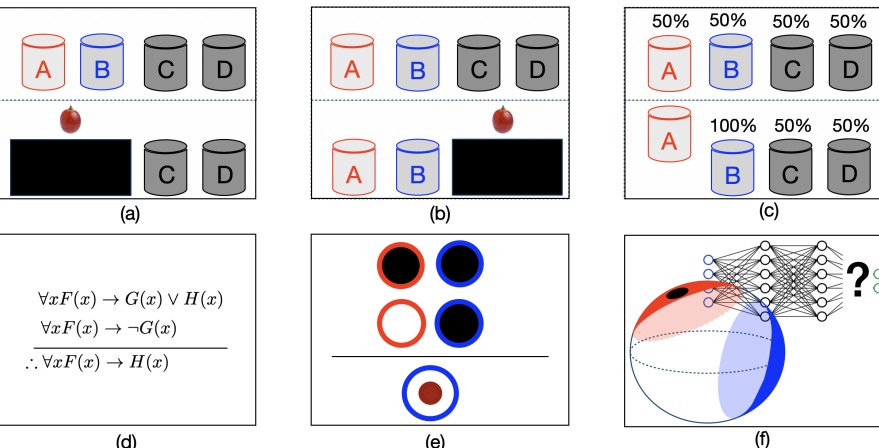

Figure 2: There are four jars A, B, C, and D in front of monkeys. (a) A blackboard covers jars A and B. A grape is dropped into one of jars A or B behind the blackboard; (b) Another blackboard covers jars C and D. Another grape is dropped into one of jars C or D behind the blackboard; (c) After removing the blackboards, monkeys will see four jars, each having 50% probability to contain a grape. If jar A is empty, clever monkeys will lift jar B. (d) The decision process of the clever monkeys can be abstracted as disjunctive syllogistic reasoning; (e) a diagrammatic representation for disjunctive syllogistic reasoning; (f) sphere neural networks and supervised neural networks may simulate disjunctive syllogistic reasoning.

pendix, we report the performance of OpenAI GPT-5 and GPT-5-nano in syllogistic reasoning). We challenge them on various kinds of syllogistic-style decision-making, which are the core of many high-stakes applications, e.g., legal judgments (Deng et al., 2023; Constant, 2023), medical diagnoses (Mushlin & Harry L. Greene, 2010; Ambags et al., 2023).

The contributions of this paper are multifold: (1) by defining Euler diagrams as circles on the surface of $n$-dimensional sphere, we enhance the representation power of Sphere Neural Networks (Dong et al., 2024; 2025) (a representative network in Category 3) for performing 16 first-order syllogistic-styled reasoning types (Betz et al., 2021), including disjunctive syllogistic reasoning; (2) We evaluated our Sphere Neural Network on constructing Euler diagrams with dimensions of 2, 3, 15, 30, 100, 200, 1000, 2000, 3000, and 10000, and it achieved 100% accuracy across all these types, showing that our Sphere Neural Network preserved the rigour of all types of classic syllogism. (3) We repurposed Euler Net (Wang et al., 2018; 2020) (a representative network in Category 1) to perform disjunctive syllogistic reasoning, and achieved 100% accuracy; (4) We created new testing data to challenge the robustness of Euler Net and demonstrated that its reasoning performance is restricted by input patterns. Our experiment results favour Category 3 networks for reliable reasoning and decision-making.

The rest of the paper is structured as follows: In Section 2, we revisit the design of Ferrigno et al. (2021)'s monkey experiments; In Section 3, we survey syllogistic-styled reasoning and some related work; In Section 4, we present our version of Sphere Neural Networks for various syllogistic reasoning; In Section 5, we conduct a comparative study between Sphere Neural Network and Euler Net and report our experiment results. In Section 6, we conclude our work and list potential future research directions.

## 2 CLEVER MONKEYS CAN DO DISJUNCTIVE SYLLOGISTIC REASONING

Ferrigno et al. (2021) conducted a series of experiments with monkeys, showing that clever monkeys can perform disjunctive syllogistic reasoning. The process can be briefly described as follows: Researchers put four jars in front of monkeys, and hide the first two jars behind a board. Then, they drop a grape into one of the two jars. Thus, the monkeys know one of the two jars has the grape, but do not know which one. Then, they repeat the process for the last two jars. Thus, from the monkeys perspective, each jar has a 50% chance of having a grape. Now, researchers lift the first jar; if it

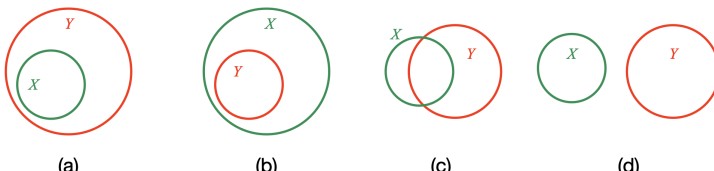

Figure 3: Basic diagrammatic representation for syllogistic statements. (1) that *all X are Y* is represented by (a) $X \subset Y$; (2) that *some X are Y* is represented by (a) $X \subset Y$ or (b) $Y \subset X$ or (c) $X \cap Y \neq \emptyset$; (3) that *no X are Y* is represented by (d) $X \cap Y = \emptyset$; (4) that *some X are not Y* is represented by (b) $Y \subset X$ or (c) $X \cap Y \neq \emptyset$ or (d) $X \cap Y = \emptyset$.

contains a grape, the monkey will get it; if not, the monkey will have the chance to lift one of the other three jars to get a grape. In this scenario, clever monkeys will lift the second jar with a $100\%$ probability of containing a grape, as illustrated in Figure 2. We may abstract the decision-making process as a form of disjunctive syllogistic reasoning as follows.

$$
\begin{array}{ll}
\begin{array}{l}
\forall x \ \text{Grape}(x) \to \text{InJarA}(x) \vee \text{InJarB}(x). \\
\forall x \ \text{Grape}(x) \to \neg\text{InJarA}(x). \\
\hline
\forall x \ \text{Grape}(x) \to \text{InJarB}(x). \quad \therefore
\end{array}
&
\begin{array}{l}
\text{gernalised} \\
\text{version:}
\end{array}
\quad
\begin{array}{l}
\forall x G(x) \to A(x) \vee B(x). \\
\forall x G(x) \to \neg A(x). \\
\hline
\forall x G(x) \to B(x). \quad \therefore
\end{array}
\end{array}
$$

## 3  SYLLOGISTIC-STYLE REASONING

Syllogistic reasoning (Jeffrey, 1981) is a form of deductive reasoning with only two premises and three terms. A very well-known example is *All Greeks are humans. All humans are mortal.* $\therefore$ *All Greeks are mortal.* The common concept *humans* in the premises establishes the relation between *Greeks* and *mortal*. Relations (or "moods" as used in the psychological literature) are limited to (1) *universal affirmative*: all $X$ are $Y$; (2) *particular affirmative*: some $X$ are $Y$; (3) *universal negative*: no $X$ are $Y$; (4) *particular negative*: some $X$ are not $Y$. By allowing terms to exchange places in the premises, we distinguish 256 different types of syllogistic reasoning (Khemlani & Johnson-Laird, 2012). The four relations can be reduced to four basic set relations in the forms of Euler diagrams (Hammer & Shin, 1998): (a) $X$ is part of $Y$ ($X \subset Y$), (b) $X$ contains $Y$ ($Y \subset X$), (c) $X$ partially overlaps with $Y$ ($X \cap Y \neq \emptyset$), and (d) $X$ is disjoint from $Y$ ($X \cap Y = \emptyset$), as shown in Figure 3. Euler diagrams can be constructed as a sphere configuration either in Euclidean space (Dong et al., 2024) or in Hyperbolic space (Dong et al., 2025) and achieve the rigour of syllogistic reasoning. The method maps Set X to Sphere $\mathcal{O}_X$ and translates four syllogistic relations into four spatial relations as follows.

- "all $X$ are $Y$" $\Leftrightarrow$ "sphere $\mathcal{O}_X$ is part of sphere $\mathcal{O}_Y$", $\mathbf{P}(\mathcal{O}_X, \mathcal{O}_Y)$;
- "some $X$ are $Y$" $\Leftrightarrow$ "sphere $\mathcal{O}_X$ does not disconnect from sphere $\mathcal{O}_Y$", $\neg\mathbf{D}(\mathcal{O}_X, \mathcal{O}_Y)$;
- "no $X$ are $Y$" $\Leftrightarrow$ "sphere $\mathcal{O}_X$ disconnects from sphere $\mathcal{O}_Y$", $\mathbf{D}(\mathcal{O}_X, \mathcal{O}_Y)$;
- "some $X$ are not $Y$" $\Leftrightarrow$ "sphere $\mathcal{O}_X$ is not part of sphere $\mathcal{O}_Y$", $\neg\mathbf{P}(\mathcal{O}_X, \mathcal{O}_Y)$.

Embedding data on a spherical surface efficiently can solve rotation-invariant learning problems (Cohen et al., 2018; Esteves et al., 2023). Here, we embed concepts as circles on the surface of an $n$-dimensional sphere and construct configurations of circles as Euler diagrams for syllogistic reasoning. If $n = 3$, these circles are cones. As an example, *Vatican is inside Italy. Italy disconnects from Greece.* $\therefore$ *Vatican disconnects from Greece.*, is illustrated visually in Figure 4(a).

When we draw a circle on the Earth's surface, the line from its centre to the circumference follows a curve along a great circle of the Earth. Formally, let $\mathcal{O}$ be an $n$-dimensional sphere with the centre $\vec{O}$ and radius $R$, $\vec{P}$ be a point on the surface of $\mathcal{O}$, $\|PO\| = R$. A circle with $\vec{P}$ as the centre and $r$ as the radius, $\bigcirc(\vec{P}, r)$, on the surface of $\mathcal{O}$, is defined as the set of points $\vec{Q}$ on the surface of $\mathcal{O}$, whose surface distance to $\vec{P}$ is less than $r$, $\arccos(\frac{\vec{P}}{\|\vec{P}\|} \cdot \frac{\vec{Q}}{\|\vec{Q}\|})R < r$. The complement of $\bigcirc(\vec{P}, r)$ is also a circle, written as $\overline{\bigcirc}(\vec{P}', r')$, where $\vec{P}' = 2\vec{O} - \vec{P}$ and $r' = (\pi - \arccos(\frac{\vec{P}}{\|\vec{P}\|} \cdot \frac{\vec{Q}}{\|\vec{Q}\|}))R$. When

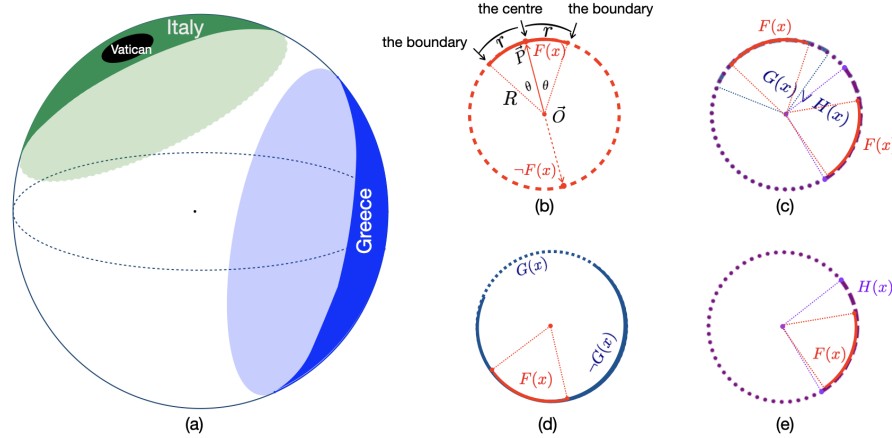

Figure 4: (a) An Euler diagram on the surface of a 3-d sphere. (b) The surface of a 2-d sphere is a circle. A sphere on this surface is a curve, e.g., $F(x)$, $\neg F(x)$; (c) $\forall x \cdot F(x) \rightarrow G(x) \vee H(x)$: the $F(x)$ arc is part of $G(x)$ arc or $H(x)$ arc; (d) $\forall x \cdot F(x) \rightarrow \neg G(x)$: $F(x)$ arc is part of $\neg G(x)$ arc; (e) $\forall x \cdot F(x) \rightarrow H(x)$: $F(x)$ arc is part of $H(x)$ arc.

$n = 2$, circles are arcs, as shown in Figure 4(b). This definition of circles is compatible with the sphere definition in (Dong et al., 2024; 2025) and can be used to create Euler diagrams for general syllogistic reasoning as follows: Let $a$ be a constant, $x$ be a variable, $F, G, H$ be predicates.

- $F(a)$ is translated into "Atomic Circle $a$ is part of Circle F", Atomic Circle $a$ has the minimal radius $\epsilon > 0$. Formally, $\mathbf{P}(\bigcirc_a, \bigcirc_F)$, where $r_a = \epsilon$.

- $\forall x F(x) \rightarrow G(x)$ is translated into "for any Atomic Circle $x$, if $x$ is part of Circle F, $x$ is part of Circle G". Formally, we write $\mathbf{P}(\bigcirc_F, \bigcirc_G)$.

- $\forall x F(x) \rightarrow G(x) \vee H(x)$. Formally, we write $\mathbf{P}(\bigcirc_F, \bigcirc_G) \vee \mathbf{P}(\bigcirc_F, \bigcirc_H)$.

- $\forall x F(x) \rightarrow \neg G(x)$. Formally, we write $\mathbf{P}(\bigcirc_F, \overline{\bigcirc_G})$.

- $\exists x F(x) \rightarrow G(x)$ is translated into "there is Atomic Circle $a$, if $a$ is part of Circle F, $a$ is part of Circle G". Formally, we write $\mathbf{P}(\bigcirc_a, \bigcirc_F) \wedge \mathbf{P}(\bigcirc_a, \bigcirc_G)$, where $r_a = \epsilon$.

We create circle configurations on an $n$-dimensional sphere for the following 16 types of syllogistic-styled reasoning in (Betz et al., 2021) (Details are described in the supplementary material).

1
$$\forall x F(x) \rightarrow G(x).$$
$$\frac{F(a).}{G(a). \quad \therefore}$$

2
$$\forall x F(x) \rightarrow \neg G(x).$$
$$\frac{F(a).}{\neg G(a). \quad \therefore}$$

3
$$\forall x F(x) \rightarrow G(x).$$
$$\frac{\neg G(a).}{\neg F(a). \quad \therefore}$$

4
$$\forall x F(x) \rightarrow \neg G(x).$$
$$\frac{G(a).}{\neg F(a). \quad \therefore}$$

5
$$\forall x F(x) \rightarrow \neg G(x).$$
$$\frac{}{\forall x G(x) \rightarrow \neg F(x). \quad \therefore}$$

6
$$\forall x F(x) \rightarrow G(x).$$
$$\frac{}{\forall x \neg G(x) \rightarrow \neg F(x). \quad \therefore}$$

7
$$\forall x F(x) \rightarrow G(x).$$
$$\forall x G(x) \rightarrow H(x).$$
$$\frac{}{\forall x F(x) \rightarrow H(x). \quad \therefore}$$

8
$$\forall x F(x) \rightarrow \neg G(x).$$
$$\forall x \neg G(x) \rightarrow H(x).$$
$$\frac{}{\forall x F(x) \rightarrow H(x). \quad \therefore}$$

9
$$\forall x F(x) \rightarrow G(x).$$
$$\forall x \neg H(x) \rightarrow \neg G(x).$$
$$\frac{}{\forall x F(x) \rightarrow H(x). \quad \therefore}$$

10
$$\forall x F(x) \rightarrow \neg G(x).$$
$$\forall x \neg H(x) \rightarrow G(x).$$
$$\frac{}{\forall x F(x) \rightarrow H(x). \quad \therefore}$$

11
$$\forall x F(x) \rightarrow G(x).$$
$$\exists x H(x) \rightarrow \neg G(x).$$
$$\frac{}{\exists x H(x) \rightarrow \neg F(x). \quad \therefore}$$

12
$$\forall x \neg F(x) \rightarrow G(x).$$
$$\exists x H(x) \rightarrow \neg G(x).$$
$$\frac{}{\exists x H(x) \rightarrow F(x). \quad \therefore}$$

13
$$\forall x F(x) \rightarrow G(x) \vee H(x).$$
$$\forall x F(x) \rightarrow \neg G(x).$$
$$\frac{}{\forall x F(x) \rightarrow H(x). \quad \therefore}$$

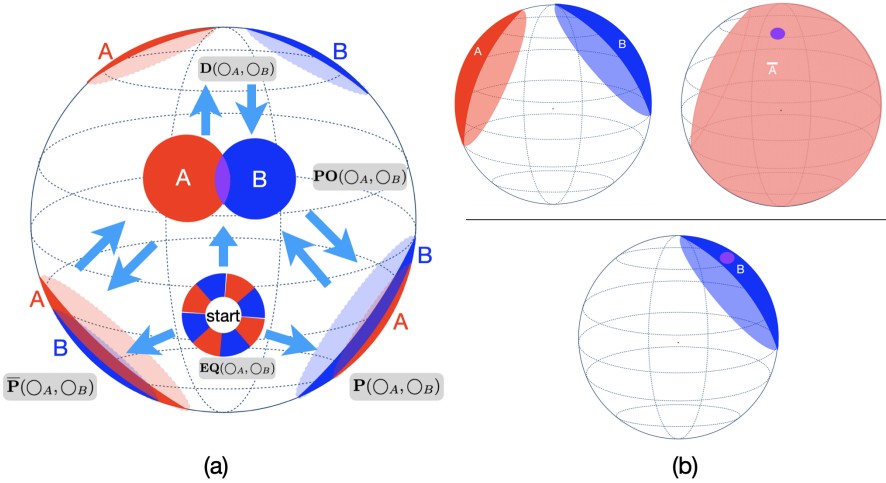

(a)             (b)

Figure 5: (a) The neighbourhood transition map between circles on the surface of a sphere. $\mathbf{PO}$ stands for "partial overlap", $\mathbf{EQ}$ stands for "equal with", $\overline{\mathbf{P}}$ stands for "inverse part of" or "contain"; (b) the circle configuration of disjunctive syllogistic reasoning: (Premise 1) a grape $\bigcirc_g$ is either in jar A or in jar B, $\mathbf{P}(\bigcirc_g, \bigcirc_A) \vee \mathbf{P}(\bigcirc_g, \bigcirc_B)$; (Premise 2) the grape is not in jar A, $\mathbf{P}(\bigcirc_g, \overline{\bigcirc_A})$; (Conclusion) the grape is in jar B, $\mathbf{P}(\bigcirc_g, \bigcirc_B)$.

| 14 | 15 | 16 |
|---|---|---|

$$\frac{\begin{array}{l}\forall x F(x) \rightarrow G(x) \vee H(x). \\ \forall x G(x) \rightarrow \neg F(x).\end{array}}{\forall x F(x) \rightarrow H(x). \quad \therefore}$$

$$\frac{\begin{array}{l}\forall x F(x) \rightarrow G(x) \vee H(x). \\ \forall x G(x) \rightarrow J(x). \\ \forall x H(x) \rightarrow J(x).\end{array}}{\forall x F(x) \rightarrow J(x). \quad \therefore}$$

$$\frac{\begin{array}{l}\forall x F(x) \rightarrow G(x) \vee H(x). \\ \forall x J(x) \rightarrow \neg G(x). \\ \forall x J(x) \rightarrow \neg H(x).\end{array}}{\forall x F(x) \rightarrow \neg J(x). \quad \therefore}$$

## 4   SPHERE NEURAL NETWORK FOR RELIABLE DISJUNCTIVE SYLLOGISTIC REASONING

Our new Sphere Neural Network (SphNN) is an extension of SphNN (Dong et al., 2024; 2025) in three aspects: (1) we define spheres as circles on an $n$-d sphere; (2) we allow complement sets and define a complement circle for the complement set; (3) our SphNN determines the satisfiability of logical formula in a disjunctive normal form $f_1 \vee ... \vee f_n$ by explicitly constructing circle configurations for syllogistic-styled statements $f_i$ one by one, where $f_i$ is a conjunctive form $g_1 \wedge ... \wedge g_m$, $g_i$ is limited to one of the forms: $\mathbf{P}(\bigcirc_X, \bigcirc_Y)$, $\neg\mathbf{P}(\bigcirc_X, \bigcirc_Y)$, $\mathbf{D}(\bigcirc_X, \bigcirc_Y)$, $\neg\mathbf{D}(\bigcirc_X, \bigcirc_Y)$, and $\mathbf{P}(\bigcirc_X, \overline{\bigcirc_Y})$.

The main process of determining the satisfiability of syllogistic statements is a gradual descent process that begins with all circles coinciding. The configuration is updated iteratively by following a neighbourhood transition map. In the setting of constructing circle configurations on the surface of a sphere, the neighbourhood transition map share the same structure as the ones in (Dong et al., 2024; 2025), with the condition that the sum of the diameters of two circles is less than the perimeter of the big circle of the sphere, as illustrated in Figure 5(a). This guarantees that our SphNN inherits the feature of *reasoning-for-sure* of the original SphNN as follows.

> *For any satisfiable syllogistic statements, SphNN can correctly construct a sphere configuration as an Euler diagram at the global loss of zero in one epoch.*

So, after the first epoch, if our SphNN fails to construct the target diagram, it will conclude that the input syllogistic statements are unsatisfiable, and the negation of the conclusion is valid (we outline the algorithm 1 in the supplementary material). To determine whether jar B contains the grape, our SphNN shall refute the assumption that jar B does not contain the grape. This is achieved by failing to construct a circle configuration for this assumption, namely, $\mathbf{P}(\bigcirc_g, \bigcirc_A) \vee \mathbf{P}(\bigcirc_g, \bigcirc_B)$,

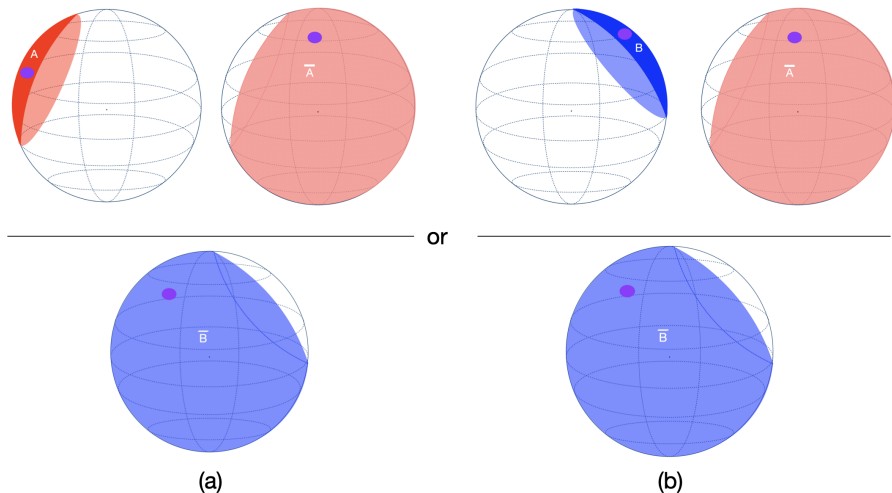

Figure 6: (a) Case 1: the grape is in jar A; (b) Case 2: the grape is in jar B. If our SphNN fails to construct a configuration for both cases, it will conclude that the grape in jar B is for sure.

$\mathbf{P}(\bigcirc_g, \overline{\bigcirc_A})$, and $\mathbf{P}(\bigcirc_g, \overline{\bigcirc_B})$. This is equivalent to two cases: (a) $\mathbf{P}(\bigcirc_g, \bigcirc_A)$, $\mathbf{P}(\bigcirc_g, \overline{\bigcirc_A})$, $\mathbf{P}(\bigcirc_g, \overline{\bigcirc_B})$, and (b) $\mathbf{P}(\bigcirc_g, \bigcirc_B)$, $\mathbf{P}(\bigcirc_g, \overline{\bigcirc_A})$, $\mathbf{P}(\bigcirc_g, \overline{\bigcirc_B})$, as illustrated in Figure 6. Our SphNN tries to construct a circle configuration for each case. After failing in both cases, it will determine for sure that the grape is in jar B.

## 5    EXPERIMENTS

We conducted four experiments to compare our SphNN and Euler Net with various training and testing datasets. In Experiment 1, we demonstrate that our SphNN achieves symbolic-level rigour in 16 syllogistic-style reasoning types (Betz et al., 2021). In Experiment 2, we show that our SphNN keeps its rigour in classic syllogistic reasoning. In Experiment 3, we retrain Euler Net (Wang et al., 2018; 2020) for disjunctive syllogistic reasoning, reaching 100% accuracy; In Experiment 4, we show that Euler Net achieving 100.00% in classic syllogistic reasoning can be trained to reach 100% accuracy in disjunctive syllogistic reasoning. However, after that, its performance will drop in classic syllogistic reasoning, and subsequently drop when patterns of input images are different from those in the training data.

### 5.1    EXPERIMENT 1

**Dataset**   We translate 16 syllogistic-styled reasoning in Section 3 into a circle configuration in the disjunctive normal form (details are described in the supplementary material), and generate three other syllogistic conclusions, totalling 64 different circle configurations. Among $16 \times 4 = 64$ types of syllogistic reasoning statements, 32 types are valid.

**Method**   To determine whether a syllogistic reasoning is valid, SphNN tries to construct a counter-example on the surface of a sphere. If successful, SphNN concludes that this syllogistic reasoning is invalid; otherwise, SphNN concludes that the original conclusion is valid.

**Setup**   The initial radius of a circle is $e^{-1}$. Three circles with radius $e^{-1}$ are randomly initialised as coinciding on the surface of a sphere with radius 1. We set the learning rate to 0.0001 and the maximum number of epochs $M = 1$. All experiments were conducted on MacBook Pro Apple M1 Max (10C CPU/24C GPU), 32 GB memory. We challenged SphNN to construct Poincaré spheres with the following dimensions 2, 3, 15, 30, 100, 200, 1000, 2000, 3000, 10000.

**Results**   SphNN successfully determined 32 valid and 32 invalid syllogistic reasoning types by constructing circle configurations with dimensions from 2 to 10000, totalling 640 reasoning tasks.

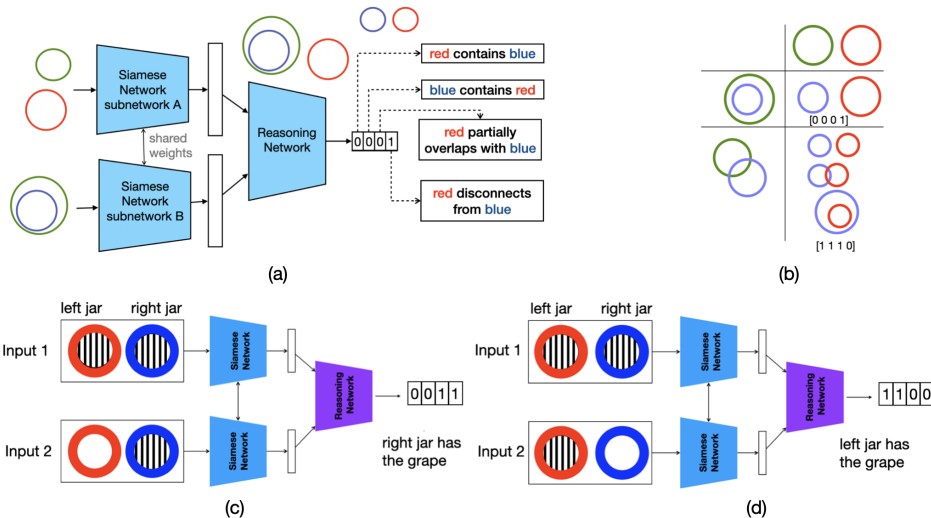

Figure 7: (a) The architecture of Euler Net; (b) The composition table of Euler Net. If the two premises are "blue is inside green. green disconnects from red", the combination result will only be "blue disconnects from red", represented as $[0, 0, 0, 1]$. (c, d) Training data of Euler Net for disjunctive syllogistic reasoning.

The mean time cost to determine a valid reasoning is 108.04 seconds. The mean time cost to construct a counter-example for invalid reasoning is 7.07 seconds; 221 among 320 (69.06%) invalid reasoning cases are determined in less than 5 seconds; 226 among 320 (70.06%) valid cases are determined in less than 120 seconds.

## 5.2 EXPERIMENT 2

**Dataset**  We use the dataset of SphNN (Dong et al., 2025).

**Method and Setup**  The same as in Experiment 1.

**Results**  Our SphNN successfully determined 24 valid and 232 invalid syllogistic reasoning types by constructing circle configurations with dimensions from 2 to 10000, totalling 2560 reasoning tasks. The mean time cost to determine a valid reasoning is 64.00 seconds. The mean time cost to construct a counter-example for invalid reasoning is 11.27 seconds; 2137 among 2320 (92.11%) invalid reasoning cases are determined in less than 5 seconds; 199 among 240 (82.92%) valid cases are determined in less than 120 seconds.

## 5.3 A COMPARATIVE STUDY WITH SUPERVISED DEEP LEARNING

Developing neural syllogistic reasoning was extremely challenging and once considered utopian (Khemlani & Johnson-Laird, 2012). Only recently were supervised neural networks, Euler Net, developed to approximate a substantial part of syllogistic reasoning (Wang et al., 2018; 2020), as illustrated in Figure 7(a). The inputs of EN are two images, each consisting of two coloured circles with a set-theoretic relation. Colours of circles distinguish three terms in syllogistic reasoning. The common colour in the two input images is the midterm. With two Siamese networks, Euler Net encodes each input image into a latent vector. The output of EN is a vector representing the set-theoretic relation(s) between the subject and the predicate. The mapping from two premises to conclusions is enumerated in the combination table, where possible conclusions are symbolised as a vector, as illustrated in Figure 7(b). The training data takes the form of ((image, image), vector). Euler Net can be trained to handle variants of classical syllogistic reasoning (Wang et al., 2018).

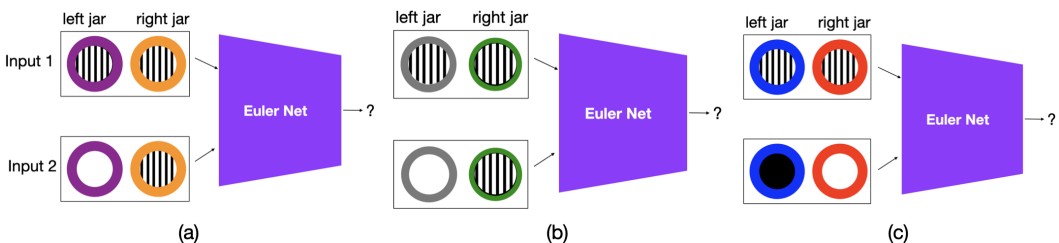

Figure 8: Three variant testing datasets challenge the well-trained Euler Net in Experiment 3.

### 5.3.1 EXPERIMENT 3

**Dataset**  We repurpose Euler Net (Wang et al., 2018) for disjunctive syllogistic reasoning. The training data consists of two premises and a conclusion vector, representing the "grape-in-jar" scenario. The first input image included two circles covered with stripes to represent uncertainty. The colour, thickness, and relative positioning of the circle were fixed, while the location of the centre was randomly varied. The second input image was identical to the first except that one circle was randomly chosen to be empty, indicating that it does not contain the grape. The output vector encoded the conclusion of the disjunctive reasoning: [1100] if the grape was in the left circle and [0011] if the grape was in the right circle, as illustrated in Figure 7(c,d).

**Setup**  Following the experiment settings in (Wang et al., 2018), we created 96000 input-output records, among which 88000 records were used for training, 8000 records were used for validation and testing.

**Results**  Euler Net achieved $100\%$ accuracy in disjunctive syllogistic reasoning on our testing dataset.

### 5.3.2 EXPERIMENT 4

We examine whether the reasoning performance of Euler Net is affected by changes in input patterns and whether Euler Net can still achieve high performance in classic syllogistic reasoning.

**Dataset**  We designed three variant input patterns for disjunctive syllogistic reasoning, as shown in Figure 8(a-c). In the first variation, the colour of the circle was randomised; In the second variation, both colour and thickness were randomised; In the third variation, the first input remained the same, but the second input depicted one circle as clear and the other as full. We generated testing data for classic syllogistic reasoning. The testing dataset for all experiments was the same as Experiment 3.

**Results**  Accuracy decreased as the visual features varied: $75.00\%$ with randomised colour, $53.57\%$ with both colour and thickness variation, and $46.43\%$ with the alternative clear/full input pattern. For classic syllogistic reasoning, its performance dropped to $6.25\%$.

## 6 CONCLUSIONS AND OUTLOOKS

Disjunctive syllogistic reasoning is a fundamental decision-making strategy that enables humans, animals, and AI systems to reach conclusions by eliminating incorrect candidates, and it has wide-ranging applications. In this work, we compare three kinds of neural networks for basic decision-making, demonstrating the advantages of the method of *reasoning through explicit model*: (1) interpretable – this method constructs human-understandable models; (2) no need for training data – this method saves many resources and eliminates biases introduced by training data; (3) extremely high performance – with this method, neural networks can achieve symbolic-level logical reasoning; (4) supporting continuous learning – new learned construction method can live with old ones. Future research directions include seeking seamless integration of SphNNs with supervised deep learning and LLMs, and applying this method for cognitive modelling and developing interpretable and safe AI models for high-stakes domains.

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

## A  CODE AND DATA

Code and data are available at: https://anonymous.4open.science/r/EN_SphNN-45DE/README.md

## B  ADDITIONAL EXPERIMENTS WITH GPT-5 FAMILY

In Section 1, we surveyed the recent evaluations of LLMs' reasoning performances in syllogistic reasoning, including disjunctive syllogism. These studies converge on the conclusion that LLMs still cannot perform perfect syllogistic reasoning. However, we may optimistically hope that in the near future LLMs will be able to. Therefore, we conducted additional experiments to find an LLM and a matching input form that can achieve 100% accuracy for classic syllogistic reasoning.

We designed four input patterns as follows, totalling $256 \times 4 = 1024$ classic syllogistic reasoning tasks.

1. Syllogistic statements with meaningful words, e.g. *all Greeks are humans*;
2. Syllogistic statements whose concepts are a string of two words connected by '_', e.g. *some Greeks_Mountain are not Humans_Tomato*;
3. Syllogistic statements with simple symbols, e.g. *some S are P*;
4. Syllogistic statements with long random symbols, e.g. *some X2asfi3f23 are aH21ü01qH*.

We let OpenAI GPT-5 and GPT-5-nano decide whether a syllogistic reasoning is satisfiable and construct an Euler diagram if possible.

The result is that GPT-5 only achieved 100% accuracy in determining the satisfiability when fed with long random symbols. However, roughly 10% of those correct decisions, GPT-5 created incorrect Euler diagrams. From GPT-5-nano to GPT-5, there were no significant improvements in limiting the number of correct decisions with wrong explanations. The training process of a neural network typically stops when it reaches 100% accuracy, which indicates the ultimate unreliability of GPT-5's decision-making. Details are listed in Table 1.

## C  CIRCLE CONFIGURATIONS FOR 16 REASONING TYPES

We listed circle configurations on the surface of a $n$-dimensional sphere as Euler diagrams for 16 syllogistic-styled reasoning types in (Betz et al., 2021) as follows.

Table 1: Syllogistic reasoning performance of OpenAI GPT-5-nano and GPT-5. 'expl' for correct explanation, 'wrg' for wrong explanation. The '#correct decision-wrg' column means a correct decision with a wrong explanation; the '#wrong decision-expl' column means a wrong decision with a correct explanation.

| version | surface form | #correct decision | | #wrong decision | | #simple acc |
|---|---|---|---|---|---|---|
| | | expl | wrg | expl | wrg | |
| gpt-5-nano | words | 160 ( 62.5%) | 90 | 5 | 1 | 97.7% |
| | double words | 230 ( 89.4%) | 22 | 4 | 0 | 98.4% |
| | simple symbols | 226 ( 88.3%) | 24 | 6 | 0 | 97.7% |
| | long random symbols | 222 ( 86.7%) | 25 | 9 | 0 | 96.5% |
| gpt-5 | words | 239 ( 93.4%) | 16 | 1 | 0 | 99.6% |
| | double words | 234 ( 91.4%) | 21 | 1 | 0 | 99.6% |
| | simple symbols | 236 ( 92.2%) | 15 | 5 | 0 | 98.0% |
| | long random symbols | 231 ( 90.2%) | 25 | 0 | 0 | **100.0%** |

## 1. Generalised modus ponens

$$\forall x F(x) \to G(x).$$
$$\frac{F(a).}{G(a). \quad \therefore}$$

Its circle configuration reads as "Circle F is part of Circle G. Atomic Circle $a$ is part of Circle F. Atomic Circle $a$ is part of Circle G.", written as follows.

$$\text{all F are G, all } a \text{ are F, all } a \text{ are G}$$

The other three syllogistic conclusions are as follows.

$$\text{all F are G, all } a \text{ are F, no } a \text{ are G}$$

$$\text{all F are G, all } a \text{ are F, some } a \text{ are G}$$

$$\text{all F are G, all } a \text{ are F, some } a \text{ are not G}$$

## 2. Negation variant of Generalised modus ponens

$$\forall x F(x) \to \neg G(x).$$
$$\frac{F(a).}{\neg G(a). \quad \therefore}$$

Its circle configuration reads as "Circle F is part of the complement Circle of Circle G. Atomic Circle $a$ is part of Circle F. Atomic Circle $a$ is part of the complement Circle of Circle G.", written as follows.

$$\text{all F are c\_G, all } a \text{ are F, all } a \text{ are c\_G}$$

We use "c\_" to represent "the complement Circle of". The other three syllogistic conclusions are as follows.

$$\text{all F are c\_G, all } a \text{ are F, no } a \text{ are c\_G}$$

$$\text{all F are c\_G, all a are F, some a are c\_G}$$

$$\text{all F are c\_G, all } a \text{ are F, some } a \text{ are not c\_G}$$

## 3. Generalised contraposition

$$\forall x F(x) \to \neg G(x).$$
$$\frac{}{\forall x G(x) \to \neg F(x). \quad \therefore}$$

Its circle configuration reads as "Circle F is part of the complement Circle of Circle G. Circle G is part of the complement Circle of Circle G.", written as follows.

$$\text{all F are c\_G, all G are c\_F}$$

The other three syllogistic conclusions are as follows.

$$\text{all F are c\_G, no G are c\_F}$$

$$\text{all F are c\_G, some G are c\_F}$$

$$\text{all F are c\_G, some G are not c\_F}$$

### 4. Negation variant of Generalised contraposition

$$\frac{\forall x F(x) \rightarrow G(x).}{\forall x \neg G(x) \rightarrow \neg F(x). \quad \therefore}$$

Its circle configuration reads as "Circle F is part of Circle G. The complement Circle of Circle F is part of the complement Circle of Circle F.", written as follows.

$$\text{all F are G, all c\_G are c\_F}$$

The other three syllogistic conclusions are as follows.

$$\text{all F are G, no c\_G are c\_F}$$

$$\text{all F are G, some c\_G are c\_F}$$

$$\text{all F are G, some c\_G are not c\_F}$$

### 5. Hypothetical syllogism 1

$$\frac{\forall x F(x) \rightarrow G(x).}{\forall x G(x) \rightarrow H(x).}$$
$$\frac{}{\forall x F(x) \rightarrow H(x). \quad \therefore}$$

Its circle configuration reads as "Circle F is part of Circle G. Circle G is part of Circle H. Circle F is part of Circle H", written as follows.

$$\text{all F are G, all G are H, all F are H}$$

The other three syllogistic conclusions are as follows.

$$\text{all F are G, all G are H, no F are H}$$

$$\text{all F are G, all G are H, some F are H}$$

$$\text{all F are G, all G are H, some F are not H}$$

### 6. Negation variant of Hypothetical syllogism 1

$$\frac{\forall x F(x) \rightarrow \neg G(x).}{\forall x \neg G(x) \rightarrow H(x).}$$
$$\frac{}{\forall x F(x) \rightarrow H(x). \quad \therefore}$$

Its circle configuration reads as "Circle F is part of Circle G. Circle G is part of Circle H. Circle F is part of Circle H", written as follows.

$$\text{all F are c\_G, all c\_G are H, all F are H}$$

The other three syllogistic conclusions are as follows.

all F are c_G, all c_G are H, no F are H

all F are c_G, all c_G are H, some F are H

all F are c_G, all c_G are H, some F are not H

**7. Hypothetical syllogism 2**

$$\forall x F(x) \to G(x).$$
$$\forall x \neg H(x) \to \neg G(x).$$
$$\overline{\forall x F(x) \to H(x). \quad \therefore}$$

Its circle configuration reads as "Circle F is part of Circle G. The complement Circle of Circle H is part of the complement Circle of Circle G. Circle F is part of Circle H", written as follows.

all F are G, all c_H are c_G, all F are H

The other three syllogistic conclusions are as follows.

all F are G, all c_H are c_G, no F are H

all F are G, all c_H are c_G, some F are H

all F are G, all c_H are c_G, some F are not H

**8. Negation variant of Hypothetical syllogism 2**

$$\forall x F(x) \to \neg G(x).$$
$$\forall x \neg H(x) \to G(x).$$
$$\overline{\forall x F(x) \to H(x). \quad \therefore}$$

Its circle configuration reads as "Circle F is part of the complement Circle of Circle G. The complement Circle of Circle H is part Circle of Circle G. Circle F is part of Circle H", written as follows.

all F are c_G, all c_H are G, all F are H

The other three syllogistic conclusions are as follows.

all F are c_G, all c_H are G, no F are H

all F are c_G, all c_H are G, some F are H

all F are c_G, all c_H are G, some F are not H

**9. Hypothetical syllogism 3**

$$\forall x F(x) \to G(x).$$
$$\exists x H(x) \to \neg G(x).$$
$$\overline{\exists x H(x) \to \neg F(x). \quad \therefore}$$

Its circle configuration reads as "Circle F is part of Circle G. An atomic Circle $a$ is part of Circle H. The atomic Circle $a$ is part of the complement Circle of Circle G. The atomic Circle $a$ is part of the complement Circle of Circle F", written as follows.

all F are G, all $a$ are H, all $a$ are c_G, all $a$ are c_F

The other three syllogistic conclusions are as follows.

all F are G, all $a$ are H, all $a$ are c_G, no $a$ are c_F

all F are G, all $a$ are H, all $a$ are c_G, some $a$ are c_F

all F are G, all $a$ are H, all $a$ are c_G, some $a$ are not c_F

**10. Negation variant of Hypothetical syllogism 3**

$$\forall x \neg F(x) \rightarrow G(x).$$
$$\exists x H(x) \rightarrow \neg G(x).$$
$$\overline{\exists x H(x) \rightarrow F(x).} \quad \therefore$$

Its circle configuration reads as "The complement Circle of Circle F is part of Circle G. An atomic Circle $a$ is part of Circle H. The atomic Circle $a$ is part of the complement Circle of Circle G. The atomic Circle $a$ is part of of Circle F", written as follows.

all c_F are G, all $a$ are H, all $a$ are c_G, all $a$ are F

The other three syllogistic conclusions are as follows.

all c_F are G, all $a$ are H, all $a$ are c_G, no $a$ are F

all c_F are G, all $a$ are H, all $a$ are c_G, some $a$ are F

all c_F are G, all $a$ are H, all $a$ are c_G, some $a$ are not F

**11. Generalised modus tollens**

$$\forall x F(x) \rightarrow G(x).$$
$$\neg G(a).$$
$$\overline{\neg F(a).} \quad \therefore$$

Its circle configuration reads as "Circle F is part of Circle G. Atomic Circle $a$ is part of the complement Circle of Circle F. Atomic Circle $a$ is part of the complement Circle of Circle F.", written as follows.

all F are G, all $a$ are c_G, all $a$ are c_F

The other three syllogistic conclusions are as follows.

all F are G, all $a$ are c_G, no $a$ are c_F

all F are G, all $a$ are c_G, some $a$ are c_F

all F are G, all $a$ are c_G, some $a$ are not c_F

**12. Negation variant of Generalised modus tollens**

$$\forall x F(x) \rightarrow \neg G(x).$$
$$G(a).$$
$$\overline{\neg F(a).} \quad \therefore$$

Its circle configuration reads as "Circle F is part of the complement Circle of Circle G. Atomic Circle $a$ is part of Circle F. Atomic Circle $a$ is part of the complement Circle of Circle F.", written as follows.

all F are c_G, all $a$ are G, all $a$ are c_F

The other three syllogistic conclusions are as follows.

all F are c_G, all $a$ are G, no $a$ are c_F

all F are c_G, all $a$ are G, some $a$ are c_F

all F are c_G, all $a$ are G, some $a$ are not c_F

**13. Disjunctive syllogism**

$$\forall x F(x) \rightarrow G(x) \vee H(x).$$
$$\forall x F(x) \rightarrow \neg G(x).$$
$$\overline{\forall x F(x) \rightarrow H(x).} \quad \therefore$$

Its circle configuration reads as "Circle F is part of Circle G or Circle H. Circle F is part of the complement Circle of Circle G. Circle F is part of Circle H.", written as follows.

all F are G_or_H, all F are c_G, all F are H

The other three syllogistic conclusions are as follows.

all F are G_or_H, all F are c_G, no F are H

all F are G_or_H, all F are c_G, some F are H

all F are G_or_H, all F are c_G, some F are not H

**14. Negation variant of Disjunctive syllogism**

$$\forall x F(x) \rightarrow G(x) \vee H(x).$$
$$\forall x G(x) \rightarrow \neg F(x).$$
$$\overline{\forall x F(x) \rightarrow H(x). \quad \therefore}$$

Its circle configuration reads as "Circle F is part of Circle G or Circle H. Circle G is part of the complement Circle of Circle F. Circle F is part of Circle H.", written as follows.

all F are G_or_H, all G are c_F, all F are H

The other three syllogistic conclusions are as follows.

all F are G_or_H, all G are c_F, no F are H

all F are G_or_H, all G are c_F, some F are H

all F are G_or_H, all G are c_F, some F are not H

**15. Generalised dilemma**

$$\forall x F(x) \rightarrow G(x) \vee H(x).$$
$$\forall x G(x) \rightarrow J(x).$$
$$\forall x H(x) \rightarrow J(x).$$
$$\overline{\forall x F(x) \rightarrow J(x). \quad \therefore}$$

Its circle configuration reads as "Circle F is part of Circle G or Circle H. Circle G is part of Circle J. Circle H is part of Circle J. Circle F is part of Circle J.", written as follows.

all F are G_or_H, all G are J, all H are J, all F are J

The other three syllogistic conclusions are as follows.

all F are G_or_H, all G are J, all H are J, no F are J

all F are G_or_H, all G are J, all H are J, some F are J

all F are G_or_H, all G are J, all H are J, some F are not J

**16. Negation variant of Generalised dilemma**

$$\forall x F(x) \rightarrow G(x) \vee H(x).$$
$$\forall x J(x) \rightarrow \neg G(x).$$
$$\forall x J(x) \rightarrow \neg H(x).$$
$$\overline{\forall x F(x) \rightarrow \neg J(x). \quad \therefore}$$

Its circle configuration reads as "Circle F is part of Circle G or Circle H. Circle G is part of the complement Circle of Circle G. Circle J is part of the complement Circle of Circle J. Circle F is part of the complement Circle of Circle J.", written as follows.

all F are G_or_H, all J are c_G, all J are c_H, all F are c_J

The other three syllogistic conclusions are as follows.

all F are G_or_H, all J are c_G, all J are c_H, no F are c_J

all F are G_or_H, all J are c_G, all J are c_H, some F are c_J

all F are G_or_H, all J are c_G, all J are c_H, some F are not c_J

## D  ALGORITHM

Given a target circle configuration, we translate it into disjunctive normal form $f_1 \vee \cdots \vee f_m$. Each $f_i$ is a conjunctive form of circle relations. For each $f_i$, we search a looped chain of circles (in the chain, either a circle or its complement circle can appear, but not both), and let the SphNN determine whether this loop configuration of circles is satisfiable. If one $f_i$ is satisfiable, our SphNN will conclude $f_1 \vee \cdots \vee f_m$ satisfiable. For example, the circle configuration $\mathbf{P}(\bigcirc_A, \overline{\bigcirc_B}), \mathbf{P}(\overline{\bigcirc_B}, \bigcirc_C)$, and $\neg\mathbf{D}(\bigcirc_A, \bigcirc_C)$ is a loop relation among Circle A, the complement Circle of Circle B, and Circle C.

---

**Algorithm 1:** The main procedure of our SphNN

**Input:** A target circle configuration on an $n$-dimensional sphere.
**Output:** The input configuration is **Satisfiable** or **Unsatisfiable**

1 Transform the input configuration into the disjunctive normal form $f_1 \vee \cdots \vee f_m$;
2 Loss $\leftarrow +\infty$;
3 **foreach** $f_i \in [f_1 \ldots f_m]$ **do**
4      CircleLoop get a loop of circles from $f_i$;
5      **if** CircleLoop $== \emptyset$ **then**
6          continue
7      **else**
8          Loss $\leftarrow$ construct a configuration for CircleLoop;
9          **if** Loss $== 0$ **then**
10             break

11 **if** Loss $== 0$ **then**
12      **return Satisfiable**
13 **else**
14      **return Unsatisfiable**

---

