# OpenReview forum: "An AI Monkey Gets Grapes for Sure -- Sphere Neural Networks for Reliable Decision-Making"
_ICLR.cc/2026/Conference — Submitted to ICLR 2026_

### Official Review · Reviewer_wZTU · 2025-10-27

**Soundness:** 2
**Presentation:** 3
**Contribution:** 2
**Rating:** 6
**Confidence:** 3

**Summary:**

This paper studies reliable neural decision-making by revisiting disjunctive syllogistic reasoning. The authors introduce an improved Sphere Neural Network (SphNN) that represents logical concepts as circles on the surface of an n-dimensional sphere, allowing explicit modeling of logical relations including negation through complementary circles. By constructing and inspecting circle configurations, the model determines the satisfiability of syllogistic statements and performs reasoning with certainty, without relying on training data. Experiments show that SphNN achieves perfect accuracy across 16 types of syllogistic reasoning tasks, outperforming supervised neural networks such as Euler Net, which show strong dependence on input patterns. Additional tests with GPT-5 models highlight the limitations of large language models in logical reasoning and emphasize the advantages of explicit model-construction methods for interpretable and reliable AI reasoning.

**Strengths:**

The paper’s strengths lie in its clear motivation, solid theoretical grounding, and comprehensive experimental validation. It tackles an important and underexplored question of how to achieve reliable decision-making in neural systems, grounding the study in both cognitive evidence and formal logic.

The proposed Sphere Neural Network introduces a conceptually elegant geometric framework that enables explicit and interpretable reasoning through spatial model construction, and the experiments are extensive and convincing, showing consistent 100% reasoning accuracy across multiple syllogistic forms and dimensions.

Overall, the work contributes a promising direction toward interpretable, data-free, and logically consistent neural reasoning, which is particularly relevant for safety-critical AI applications.

**Weaknesses:**

While the paper makes a strong conceptual and empirical contribution, it also has several limitations.

1. The paper focuses on syllogistic-style reasoning tasks that are intentionally idealized to allow for precise theoretical and geometric analysis. While this setting is appropriate for validating the rigor and interpretability of the proposed model, it also limits the scope of evaluation. The study does not yet demonstrate how the method performs under more complex or noisy reasoning scenarios that involve uncertain information, contextual ambiguity, or real-world data.

2. The evaluation mainly focuses on logical correctness and accuracy, but it offers limited discussion on robustness or computational efficiency compared with other reasoning paradigms such as neuro-symbolic or constraint-based approaches. For instance, the paper reports the mean time cost but lacks comparative runtime analyses. A more systematic benchmarking study would help clarify the practical trade-offs and scalability of the proposed method.

**Questions:**

1. In my view, the paper mainly focuses on syllogistic-style reasoning in idealized settings. How do the authors envision applying the proposed method to real-world decision-making tasks? What kinds of practical problems could this framework potentially address?

2. About runtime:
- Although the experiments have evaluated different n values up to 10,000, it remains unclear how the model’s runtime and efficiency change as the reasoning tasks become larger or more complex. Could the authors provide a clearer analysis of how scalability affects runtime and what adjustments, if any, are needed to maintain efficiency?
- Could the authors include a more detailed efficiency comparison against other baseline methods?

---

> ### Author Response · Authors · 2025-11-16
>
> 1. What kinds of practical problems could this framework potentially address?
>
> Thank you for your insightful comments.
> The proposed method solves reasoning tasks by constructing sphere configurations as set-theoretic models in a vector space and attains the rigour of symbolic-level syllogistic reasoning.
> The centre of a sphere is a vector. Thus, sphere embeddings do not exclude data-driven neural networks from predicting sphere centres if training data are available. The non-zero radius of the sphere embedding introduces the part-whole relation between spheres and builds up set-theoretic semantics, which is missing in current data-driven neural networks.
>
> The set-theoretic relation is the foundation of almost every discipline and has wide applications. Syllogistic reasoning is the start of the history of logical reasoning, focusing on four basic relations between sets. Syllogistic reasoning is ubiquitous, from primary school education to business advertising, legal judgments, and biomedicine [1-5].
>
> [1] D. Ardelean, How do primary school pupils think? syllogistic reasoning in primary school children, Procedia  Social and Behavioral Sciences 203 (2015) 57–62, international Conference EPC-TKS 2015.
>
> [2] P. Jaganathan, S. Mayr, F. Nagaratnam, Rhetorical syllogism in the English and the German language of automobile advertising, Gema Online Journal of Language Studies 151 (2014) 151–169.
>
> [3] W. Deng, J. Pei, K. Kong, Z. Chen, F. Wei, Y. Li, Z. Ren, Z. Chen, P. Ren (2023), Syllo gistic reasoning for legal judgment analysis, EMNLP.
>
> [4] A. Constant, A Bayesian model of legal syllogistic reasoning, Artificial Intelligence and Law 32 (2023) 1–22.
>
> [5] Magdalena Wysocka, Danilo Carvalho, Oskar Wysocki, Marco Valentino, and Andre Freitas. SylloBio-NLI: Evaluating large language models on biomedical syllogistic reasoning.
>
> 2. How do the authors envision applying the proposed method to real-world decision-making tasks?
>
> Applying the proposed method to real-world decision-making tasks is a hot ongoing task. One possibility is to construct sphere embeddings of a symbolic taxonomy in the embedding space, while preserving the pre-trained vectors as sphere centres. This brings the hierarchy of taxonomy explicitly into the embedding space and can greatly improve the precision and the interpretability of data-driven classification methods [6].
>
> [6] Tiansi Dong and Rafet Sifa (2024). Word Sense Disambiguation as a Game of Neurosymbolic Darts. In: LREC-COLING Workshop “Bridging Neurons and Symbols for Natural Language Processing and Knowledge Graphs Reasoning”  https://aclanthology.org/2024.neusymbridge-1.3/
>
> 3. Could the authors include a more detailed efficiency comparison against other baseline methods?
>
> Efficiency is a critical issue for real applications. At the README.md of the github repo (in the appendix), we listed the detailed efficiency of SphNN and EulerNet in determining the validity of syllogistic statements.
> https://anonymous.4open.science/r/EN_SphNN-45DE/README.md SphNN took more time to determine a valid syllogistic reasoning being valid than an invalid syllogistic reasoning being invalid. The maxium time for SphNN to determine “a valid syllogism being valid” is 4002.25 seconds; The maxium time for SphNN to determine “an invalid syllogism being invalid” is 332.25 seconds. The average time for SphNN to determine “a valid syllogism being valid” is 108.04 seconds; The average time for SphNN to determine “an invalid syllogism being invalid” is 7.07 seconds. This difference does not appear for EulerNet. Its average response time is 0.025 seconds – much faster than SphNN. We will include a more detailed efficiency comparison in the final version.
>
> We will include all your comments in the final version and in the appendix.

---

### Official Review · Reviewer_GPLZ · 2025-10-29

**Soundness:** 3
**Presentation:** 2
**Contribution:** 2
**Rating:** 2
**Confidence:** 3

**Summary:**

This paper proposes Sphere Neural Networks (SphNN) for logical reasoning. SphNN reason by explicitly constructing geometric models. It embeds concepts as circles on an n-dimensional sphere and represent the negation operation via "complement circles." Experiments show that it can rigorously solve 16 types of syllogistic reasoning with 100% accuracy. In a comparative study, a supervised Euler Net is shown to be vulnerable to input distribution shift. After being retrained for a new task, its accuracy on classic syllogisms drops significantly. The paper concludes that this model-construction approach is superior for achieving reliable and robust reasoning and decision-making.

**Strengths:**

1.	The paper focuses on an important problem, i.e., how to let a model conduct logical reasoning in a deterministic and reliable way.

2.	The proposed improvement over the original SphNN is novel and the illustrations in figures are clear.

**Weaknesses:**

1.	The model is restricted to highly structured input, e.g., syllogistic reasoning tasks. It is unclear how the proposed method would scale to more complex, real-world reasoning that involves ambiguity, common sense, or relational logic (e.g., "A is taller than B, B is to the left of C"). Moreover, the paper assumes that logical statements are already perfectly translated into formal representations. In reality, reasoning problems may come in the form of ambiguous natural language.

2.	The time complexity of the proposed method is not carefully discussed. The proposed method seems to deal with disjunction by testing each possibility. This is feasible for simple cases but would yield a high computational cost with many disjunctive clauses.

3.	The paper claims that there is “no need for training data.” This is technically correct for the proposed method. However, the model itself (including algorithms and transition rules) is entirely designed by human experts. This simply shifts the burden from data collection to complex, expert-driven algorithm design. The generalizability to more complex and non-structured input is also questionable.

4.	The paper demonstrates that a retrained Euler Net suffers from catastrophic forgetting. This comparative experiment is not convincing due to the following reasons. (1) The proposed SphNN cannot handle visual inputs (including corrupted ones) as Euler Net does, making this comparison somehow meaningless. It is more of showing “what Euler Net cannot do,” but not “what SphNN can do.” (2) The paper does not compare against modern continual learning techniques designed to mitigate catastrophic forgetting.

5.	Minor. The name of the proposed method “SphNN” is identical to the previous one in (Dong et al. 2024). It is better consider using a different name.

**Questions:**

1.	The paper suggests integrating SphNN with LLMs as a future direction. Are there preliminary thoughts on this direction?

Please see Weaknesses for other concerns.

---

> ### Author Response · Authors · 2025-11-16
>
> Thank you for taking the time to review this paper. There are a few misunderstandings we would like to clarify:
> (1) This paper does not propose Sphere Neural Networks for logical reasoning. SphNN was introduced in (Dong et al., 2024; 2025);
>
> (2) The idea of reasoning by model construction originates from psychological research as a key mechanism underlying rational thinking;
>
> (3) The aim of this paper is to compare three neural reasoning methods and to advocate for reasoning by explicit model construction.
>
> 1. The paper suggests integrating SphNN with LLMs as a future direction. Are there preliminary thoughts?
>
> Yes. If we treat vector embeddings as zero-radius spheres, then SphNN naturally generalizes traditional neural networks. LLMs (or other neural networks) can be used to predict the centers of spheres, while SphNN can then construct sphere configurations to perform explicit reasoning.  (Dong, 2024) has already reported an experiment where SphNN speeds up sphere configuration by using pre-trained ChatGPT vectors as orientations for sphere centers.
>
> 2. It is unclear how the proposed method would scale to more complex, real-world reasoning that ... (e.g., "A is taller than B, B is to the left of C").
>
> This is an excellent and important concern for applying AI/neural networks to real-world reasoning. Our position is that set-theoretic relations provide a more realistic foundation than similarity relations between vectors, which can be represented as part–whole relations between regions. Many relational concepts can be reduced to the connection relation between regions, a tradition in philosophy and psychology dating back to Alfred North Whitehead, Theodore de Laguna, Jean Piaget, Barry Smith and Susan Carey.
>
> “A is taller than B.” can be defined as follows: Let X be a region above ground level. (i) A connects to X while standing on the ground; (ii) B can not connect to X while standing on the ground.
>
> Orientation can be reduced to distance comparisons. The basic meaning of “left” comes from proximity to the heart.  “B is to the left of C” can be interpreted as follows:  (i) If you project your body toward C; (ii) B is closer to the side of your body where the heart is located. This reduction is formally developed in:
>
> [1] Dong, T. (2008). A Comment on RCC: from RCC to RCC++. JPL, 37(4).
>
> [2] Dong, T. & Guesgen, W. H. (2008). A Uniform Framework for Orientation Relations Based on Distance Comparison. 7th IEEE Conference on Cognitive Informatics.
>
> We can introduce fuzzy sphere boundaries to allow SphNN to represent graded membership and vague concepts.
>
> 3. The time complexity of the proposed method is not carefully discussed.
>
> The proposed method modifies only sphere shapes while keeping the underlying reasoning procedure unchanged. This does not alter the time complexity of the original method. All time costs are reported in the project’s README.md (in the appendix). https://anonymous.4open.science/r/EN_SphNN-45DE/README.md
>
> 4. The model itself is entirely designed by human experts. This simply shifts the burden from data collection to complex, expert-driven algorithm design. ...
>
> It is important to note that data-driven neural networks also rely heavily on expert-designed algorithms. Backpropagation, LSTMs, Transformers, and LLM architectures are all human-engineered frameworks. Sphere embeddings simply generalize vectors by allowing non-zero radii, enabling the representation of set-theoretic relations. This is a natural extension of traditional NN, not a departure from them. When data are available, traditional NN may optimize sphere centers.
>
> 5. The paper demonstrates that a retrained Euler Net suffers from catastrophic forgetting. This comparative experiment is not convincing ...
>
> Our goal in this comparison is not to show how Euler Net can be improved through continual learning, but rather to highlight that SphNN does not suffer from catastrophic forgetting at all.  Euler Net reasons at the surface level (symbols or images), which makes it vulnerable to new visual patterns; while SphNN reasons at the semantic level, via sphere configurations as set-theoretic semantics. Because the reasoning rules are explicit and symbolic in nature, new tasks do not overwrite previous ones. From this perspective, the comparison illustrates a methodological argument: semantic-level reasoning avoids problems inherent to syntax-level neural reasoning. We will clarify this position to avoid any misunderstanding.
>
> 5.  The name of the proposed method “SphNN” is identical to the previous one in (Dong et al. 2024).
>
> We agree this is an important issue and have contacted the authors of (Dong et al. 2024) and have received confirmation that the name “SphNN” may be reused in this context.
>
> Thank you again for your critical but constructive comments. Some issues are indeed not clearly addressed in the current version. We will include all your comments in the final version and in the appendix.

---

### Official Review · Reviewer_6LNb · 2025-10-31

**Soundness:** 2
**Presentation:** 3
**Contribution:** 2
**Rating:** 4
**Confidence:** 3

**Summary:**

The paper proposes a new version of Sphere Neural Networks that embeds concepts as circles on the surface of an n-dimensional sphere. The authors evaluate the method across 16 syllogistic-style reasoning types (and several dimensions up to 10,000), report perfect accuracy for SphNN on these tasks, compare with a retrained Euler Net supervised model, measure run-times, and include experiments with GPT-5 family as an additional comparison.

**Strengths:**

1.	Clear, structured mapping from logical relations to geometric constraints and a catalogue of 16 syllogistic types.
2.	Direct comparative experiments with Euler Net and GPT-5.
3.	Excellent and detailed drawing and presentation.

**Weaknesses:**

1.	Novelty concern: The paper does not clearly and rigorously state what technical advances are new vs prior work (Dong et al. 2024/2025). The paper devotes considerable space to the story's background and the monkey example, but fails to sufficiently highlight its own innovative points, leaving its unique aspects under-emphasized.
2.	Evaluation limitation. All evaluations are conducted through textual descriptions. Presenting the improvements achieved by the method in tabular form would be more intuitive.

**Questions:**

1.	Previous work has mentioned that HSphNN(Dong et al. 2025) can help improve GPT-3.5's performance. Does the method presented in this paper have similar applications?
2.	Large parts of the formulation, the neighbourhood transition map, and the “construct-by-descent” algorithm closely mirror the earlier SphNN work and the same research program. Could the paper more clearly delineate what is truly new and better justify its novelty beyond incremental representational tweaks?

---

> ### Author Response · Authors · 2025-11-16
>
> Thank you very much for your careful reading and constructive comments. This paper has two targets:
>
> (1) By comparing three types of neural networks on syllogistic reasoning tasks, we demonstrate several advantages of Recent Sphere Neural Networks, which perform reasoning by explicit model construction without training data. These advantages include: they are not black boxes; they are not data-hungry; they do not require training data, and therefore do not suffer from out-of-distribution issues or hallucinations; they are deterministic and reliable; they remain compatible with the two other neural-network types (via sphere embeddings with zero radius).
>
> (2) We highlight the methodological contribution of neural reasoning through explicit model construction without training data. This approach begins with basic reasoning tasks and performs reasoning by explicitly constructing set-theoretic semantics in vector space. Aristotelian syllogistic reasoning, as the earliest and most fundamental form of logical reasoning, provides a natural starting point. Our method solves reasoning tasks by transforming spheres into other geometric shapes. Technically, we search for minimal shape transformations that yield significant improvements in reasoning performance. This constitutes the core novelty of neural reasoning through explicit model construction without training data.
>
> 1. Novelty concern: Could the paper more clearly delineate what is truly new and better justify its novelty beyond incremental representational tweaks?
>
> Thanks for this constructive suggestion! We will revise the paper to more clearly articulate the methodological novelty of neural reasoning through explicit model construction without training data and to explicitly show how defining spheres as circles on a higher-dimensional sphere enhances reasoning power.
>
> 2. Previous work has mentioned that HSphNN(Dong et al. 2025) can help improve GPT-3.5's performance. Does the method presented in this paper have similar applications?
>
> Yes. HSphNN can verify the satisfiability of GPT-3.5’s syllogistic-reasoning outputs by attempting to construct an Euler diagram explicitly. When HSphNN detects an incorrect reasoning pattern, it provides targeted feedback through prompts, thereby improving GPT-3.5’s performance.
> Similarly, our SphNN can improve GPT-3.5 (as well as other LLMs) across 16 reasoning tasks plus classical syllogistic reasoning, using the same principle of explicit model construction.

---

> > ### Comment · Reviewer_6LNb · 2025-11-27
> >
> > I thank the authors' responses. It sounds OK to me. I decide to raise my score.

---

### Official Review · Reviewer_ZjM7 · 2025-11-03

**Soundness:** 2
**Presentation:** 1
**Contribution:** 2
**Rating:** 2
**Confidence:** 2

**Summary:**

he paper proposes a custom neural network architecture for solving syllogistic reasoning problems. The architecture works by using a particular geometric formulation of logic in which propositions are represented as circles/spheres which indicate their meaning in set theoretic terms. This architecture is a generalization of a previously-introduced model. The authors test performance on a number of standard syllogistic patterns and show it performs well.

My primary concern with the paper is that it is difficult to follow and pitched to a fairly niche audience, which will make it unlikely to resonate with ICLR broadly. I recommend the paper be rewritten for a more general audience before publishing at ICLR. Alternatively, the authors might consider venues that have specific interest in these topics, e.g., IWCS or similar venues.

**Strengths:**

* The problem of if/how neural networks can account for classical logical reasoning is long standing and thus novel insights and architectures that speak to this are interesting
* The focus on this particular geometry of an embedding space is, from my understanding, fairly novel

**Weaknesses:**

My primary concern with the paper is that I found it hard to follow. I am not an expert in this particular type of architecture, but do consider myself to be reasonable well versed in logic and the application of NNs to classical logic and semantics problems (at least as much as the average ICLR attendee if not moreso). Therefore, if the paper was not "clicking" for me, I take it as a signal that it needs to be reworked in order to have impact. It seems that there is a real contribution here, but I admit I can't fully articulate what it is, because much of the background and intuition was missing from the writeup. I give some more specific questions below, but my general feeling is it should undergo a round of revision before publishing at ICLR or similar venues.

**Questions:**

* Can you provide more background on Euler net? I think it is fair to assume many readers won't have seen that prior work. In particular, can you offer more intuition for why the inputs are represented as diagrams? Why not in some generic symbolic language instead? Is the image representation necessary for the architecture to work, or is that just a specific design choice of prior work? This was confusing to me, as it seems that a logical reasoning engine should be agnostic to the format of the input, and rather operate in a more abstract space.
* Can you say more (in the main text) about how the models are trained? What type of data do they train on and what are the train/test splits like? It is easy for a neural net to overfit surface patterns of syllogisms and thus to appear to perform well without internalizing any deep reasoning.
* How does this architecture perform on tasks that aren't syllogistic reasoning? E.g., language modeling? It is important to report such things, as there isn't really a point of having an NN that can do formal logic if that is _all_ it can do--the appeal of logic-capable neural networks is that they can do logic but still do other more ``neural net-y'' things too. Otherwise, we'd just use a symbolic reasoning engine to do the logic.


Typos:
* cite in first paragraph should be to mitchell, not melanie
* line 41 Gvery instead of Every
* typo in this line: Sufficient empirical experiments advocate the model theory for reasoning that reasoning is a process of constructing and inspecting mental models
* In general, there are frequent typos and oddly-worded sentences, so I stopped taking note. Make sure it gets a careful proof read on resubmission

---

> ### Author Response · Authors · 2025-11-16
>
> Thank you for your questions. We will improve the writing and make it more readable.
>
> 1. Can you provide more background on Euler net? can you offer more intuition for why the inputs are represented as diagrams? Why not in some generic symbolic language instead?
>
> Euler Net is a type of supervised neural network (NN) designed to perform reasoning with diagrammatic inputs. As illustrated in Figure 7(a), it takes two input diagrams, represented as matrices of pixels, and transforms them into latent vector representations. The model was first introduced in 2018 to approximate syllogistic reasoning and later extended in a 2021 ICLR publication to approximate abstract diagrammatic reasoning.
>
> The syllogistic statements that we are evaluating require reasoning through set-theoretic relations, which can be represented through Venn diagrams or Euler diagrams. If we use a supervised NN, its inputs will be set-theoretic diagrams of the premises, and its output will be a set-theoretic diagram of the conclusion.
>
> The motivation for using diagrammatic representations stems from cognitive and comparative reasoning studies. Many animals can reason about spatial relations without language, and human infants demonstrate relational reasoning abilities before developing speech. Even non-human primates are capable of basic syllogistic reasoning. This suggests that spatial or diagrammatic forms of reasoning may underlie symbolic reasoning in humans.
>
> Thus, we are looking for neural networks with diagrammatic inputs. Our three primary questions are the following: which neural network can perform syllogistic reasoning in diagrammatic inputs instead of symbolic language? How can this neural network reach the symbolic level of syllogistic reasoning? How can this neural network be developed into more general logical reasoning?
>
> 2. Is the image representation necessary for the architecture to work, or is that just a specific design choice of prior work?
>
> The image-based (diagrammatic) representation is not strictly necessary for the architecture to function, but it reflects a deliberate design choice aligned with our objective to develop a neural architecture capable of reasoning without relying on symbolic language. Diagrammatic representations were therefore chosen because they are a natural and widely used medium for expressing logical and set-theoretic relations.
>
> 3. Can you say more (in the main text) about how the models are trained? What type of data do they train on and what are the train/test splits like?
>
> This is a misunderstanding that we shall clarify in the paper. The Sphere Neural Network is not a trainable model in the conventional sense as it does not rely on data-driven learning or parameter optimization. Instead, it functions as a reasoning mechanism that evaluates the satisfiability of syllogistic statements geometrically. Given a set of premises and a conclusion, the model constructs a sphere configuration that represents the semantic relationships among the statements.
> If such a configuration can be successfully formed, the syllogistic statements are considered satisfiable; if not, they are unsatisfiable. In this sense, the Sphere Neural Network performs geometric reasoning rather than statistical learning, and thus notions such as training data or train/test splits do not apply.
>
> 4. How does this architecture perform on tasks that aren't syllogistic reasoning? language modeling?
>
> This is an important and insightful question. In the Sphere Neural Network, the foundational relations are set-theoretic (such as part–whole and member–of relationships). To extend the architecture beyond syllogistic reasoning, such as for language modeling, we would first represent the set-theoretic semantics of the target language and then translate these into corresponding geometric configurations. The Sphere Neural Network can then reason about the satisfiability of the resulting set of statements.
>
> The advantage of set-theoretic relations is their interpribilities, as they provide a well-established semantic framework across almost all scientific disciplines. Furthermore, the model can integrate vector embeddings from conventional supervised neural networks by assigning these embeddings to sphere centers. This hybrid approach allows the Sphere Neural Network to combine logical reasoning with approximate, data-driven predictions. Using pre-trained embeddings as sphere centers can also greatly accelerate the construction of reasoning models, as discussed in Dong et al. (2024/2025).
>
> This integration offers a promising direction for the ICLR community, which continues to seek ways to bridge black-box neural models and symbolic logic. Rather than relying solely on larger datasets, we may instead enrich latent feature vectors by introducing a radius, thus creating a sphere-based representation with higher interoperability.

---

### Meta-Review · Area_Chair_nNFQ · 2026-01-07

**Summary:**

The reviewers all saw some good ideas here, especially about making AI reasoning more reliable and interpretable by building explicit models. But most of them had significant doubts. The major concerns were that the new technical contribution seemed somewhat limited compared to the earlier SphNN papers, and that the experiments only utilized clean, simple logic puzzles. Some also found the writing too specialized and difficult to follow for a general audience, and a few felt that the comparison with another model, Euler Net, wasn't entirely fair.

**Reviewer Concerns:**

The authors did a good job in their response explaining some questions. They clarified why they set up the comparison the way they did, shared more details about how fast the system runs, and gave some ideas for how it could connect to large language models or real uses.

But the main concerns are still there. The paper still feels more like an extension than a major new step forward. It still doesn't show the method working on messy, real-world problems, which is a key test. And the writing is still aimed at a narrow group of experts.

**Reviewer Scores:**

ZjM7 (score 2): Their main issue was clarity and narrow focus. The response didn't fully fix that, so their score probably stays at 2.

6LNb (score raised to 4): They were happier after the novelty was clarified. Might have settled at a weak 5.

GPLZ (score 2): They had strong concerns about the experiment fairness and scope. I don't think the response changed their mind, so likely still 2.

wZTU (score 6): They liked the theory but wanted more proof of real-world use. Without new evidence, they might have lowered to a 5.

---

### Decision · Program_Chairs · 2026-01-26

Reject